# Improving Sleep Quality to Prevent Perinatal Depression: The Obstetric Nurse Intervention

Ana Filipa Poeira [1],* and Maria Otília Zangão [2]

1 Instituto Politécnico de Setúbal, ESS, NURSE'IN-UIESI, Campus do IPS Estefanilha, 2914-503 Setúbal, Portugal
2 Comprehensive Health Research Centre, Department of Nursing, Higher School of Nursing, University of Évora, 7000-811 Évora, Portugal
* Correspondence: ana.poeira@ess.ips.pt

**Abstract:** Throughout their life, women should pay attention to their mental health. Evidence indicates that poor sleep quality is related to depressive symptoms in pregnancy, justifying the intervention of health professionals in improving sleep quality to promote the mental health of pregnant women. The objective of our study is to analyze the relationship between sleep quality and perinatal depression, and to identify the obstetric nurse's intervention in improving sleep quality in the perinatal period. A total of 53 pregnant women between the 28th week of pregnancy and the 7th day after delivery completed the Edinburgh Postnatal Depression Scale (EPDS) and Pittsburgh Sleep Quality Index (PSQI). Women were also asked about the strategies used by the obstetric nurse to improve their quality of sleep. Data analysis was performed using IBM SPSS Statistics software, version 25.0. The Mann–Whitney-U and Kruskal-Wallis tests were carried out. A $p$-value < 0.05 was considered statistically significant. The median PSQI score was 10 ($\pm$3.63), and 9.2% ($n$ = 9) had good quality sleep. The median EPDS score was 12 ($\pm$4.43), and 27 participants (50.9%) had probable depression. The women with likely depression had worse sleep quality ($p$ = 0.016). Most participants reported that the obstetric nurse showed no interest in their sleep quality during pregnancy. Women of other nationalities have a higher risk of depression ($p$ = 0.013). Based on our results, it is crucial to assess sleep quality in the perinatal period to promote women's health during the prenatal and postnatal periods, and more action is needed since we are facing one of the most significant challenges of this century, preventing depression.

**Keywords:** health promotion; mental health; midwifery; nursing; perinatal care; postpartum depression; pregnant women; sleep hygiene

## 1. Introduction

Throughout life, a woman must pay attention to her mental health, and the quality of sleep is one of the factors needing attention. The Diagnostic and Statistical Manual of Mental Disorders (DSM-5) says that peripartum depression is depression that occurs during pregnancy or within the first four weeks postpartum [1]. This psychic disorder is not considered specific to this phase of a woman's life but is instead associated with it [1,2]. However, pregnancy, childbirth, and postpartum can potentially affect the appearance of this disorder due to hormonal, physical, physiological, and behavioral changes and the need for readjustment to the new family dynamics. According to a meta-regression that included 96 studies, perinatal depression has an estimated prevalence of 11.9% (95% CI, 11.4–12.5) [3]. Many cases of postpartum depression likely develop during pregnancy [4]. Several factors contribute to perinatal depression, and sleep quality can be associated with an increased risk of perinatal depression [5–8]. Postpartum depression is a public health problem and is associated with higher levels of anxiety and a significant number of other psychiatric symptoms: lower maternal self-esteem and less confidence in the performance of the parental role, more physical health problems, more self-injurious behaviors, and less self-care (e.g., less adherence to medical treatments, self-medication, and inadequate nutrition) [9]. The perceived level of

support from the healthcare staff during childbirth was found to be a protective factor against the development of postpartum depressive symptoms [9].

Sleep quality worsens during the third trimester of pregnancy [2,10–13]. High amounts of cortical awakenings and arousals in pregnant women cause sleep fragmentation, with more significant parts of light sleep and less deep sleep [2]. In postpartum, staying asleep is more difficult because of decreases in estrogen and progesterone levels [14]. Evidence consistently indicated that disturbances in sleep during pregnancy were related to adverse health outcomes [15], such as gestational diabetes mellitus [16] or preterm birth and low birth weight [17]. According to the study by Okun et al., they concluded that for each point of increase in the assessment of sleep quality using the Pittsburgh Sleep Quality Index (PSQI), the chances of premature birth increased by 25% in early pregnancy and 18% in later pregnancy [18]. In the study by Gao et al., they suggested that participants with poor sleep quality are more likely to suffer from stress during pregnancy, prenatal depression, and postpartum depression compared to those with good sleep quality [13].

During regular antenatal care, signs of insomnia, difficulty managing mental stress, and excessive preoccupation with pregnancy should be screened to decrease the rates of clinical depression and suicidal ideation [19]. The evaluation of sleep quality should be carried out in prenatal and postnatal routine care because it is necessary to promote women's mental health in the prenatal and postnatal periods [20]. In terms of health and implementing good practices in detecting signs and situations of risk, it is essential to emphasize the relevance of interventions that promote mental health in pregnancy. The obstetric nurse has this responsibility and competence of action, being one of the health professionals closest to the pregnant woman and family.

Despite all the existing evidence on the risks and complications associated with poor sleep quality during pregnancy, it remains relevant to understand whether there is an intervention by obstetric nurses and which intervention to prevent and improve sleep disorders in pregnant women. Promoting health and health education is an autonomous intervention and the responsibility of obstetric nurses, requiring a nursing care plan aimed at the need for sleep and rest in pregnant and postpartum women.

This study aims to analyze the relationship between sleep quality and perinatal depression and to identify the obstetric nurse's intervention in improving sleep quality in the perinatal period. It is also essential to understand what strategies have been proposed to women to improve their sleep quality and, consequently, prevent complications in their mental health. Only then will it be possible to understand what is being done and what could be suggested as strategies to implement in the scope of health promotion carried out by obstetric nurses.

## 2. Materials and Methods

This study is part of the quantitative research paradigm of descriptive and cross-sectional type. The Hospital's Ethics Committee approved the study (approval number: 47/2021), which was conducted in compliance with the principles of the Helsinki Declaration. Data collection, based on a questionnaire, was carried out between December 2021 and March 2022. An online questionnaire was applied using the Google Forms tool and sent by email. A total of 78 pregnant women who met the inclusion criteria and attended the pregnant women's consultation in a Lisbon region hospital were selected. Inclusion criteria were cumulative: Pregnant women [1] between the 28th week of pregnancy and the 7th day after delivery; [2] who understand and can express themselves in Portuguese; [3] who have access to and can use a computer or smartphone with an internet connection. Pregnant women under 18 were excluded from the study as they were legally considered minors. Of the 78 questionnaires sent, 21 were not answered, and four were excluded for not answering all questions of the instruments applied. The sample consisted of 53 pregnant women between the 28th week of pregnancy and the 7th day after delivery. Informed consent was obtained by asking all participants to select "yes" or "no" at the beginning of the online survey to consent to participate. Authorizations were requested from the authors of the scales used in this study.

## 2.1. Instruments

Sleep quality was assessed subjectively using the PSQI [21], an instrument validated for the Portuguese population [22]. The PSQI consists of 19 questions about sleep quality during the past month that are categorized into seven components: subjective sleep quality, sleep latency, sleep duration, sleep efficiency, sleep disturbances, use of sleep medication, daytime sleepiness, and dysfunction [22]. The sum of the seven components indicates the overall value of sleep quality, ranging from 0 to 21, with less than or equal to 5 indicating good sleep quality and more than 5 points indicating poor sleep quality [22]. In the present study, the 7-component score of the PSQI-PT showed an overall reliability coefficient (Cronbach's α) of 0.717, indicating an adequate degree of internal consistency.

The obstetric nurse's intervention was evaluated using a questionnaire designed by the authors for this study and consisting of three closed questions: [1] During pregnancy, did the obstetric nurse show interest in your sleep pattern at any time; [2] During pregnancy, did the obstetric nurse show interest in your sleep quality at any time; [3] List the strategies the obstetric nurse addressed to improve your sleep quality. The construction of this part of the questionnaire was supported by evidence [8,20,23–25] to list different non-pharmacological strategies for obstetric nurses in the field of intervention.

To determine the incidence of pregnant women with depressive symptoms between the 28th week of pregnancy and the 7th day after delivery, the Edinburgh Postpartum Depression Scale EPDS was applied [26], translated, and validated into Portuguese [27]. This instrument consists of 10 questions scored from 0 to 3 according to the presence or intensity of symptoms. A score of 12 or more indicates the likelihood of depression but not its severity [27]. Note that the EPDS is designed to complement, not replace, the clinical evaluation [27].

## 2.2. Statistical Treatment

The statistical treatment was performed using the IBM SPSS Statistics software, version 25.0. Descriptive statistics were used to characterize the participants, assess the quality of sleep, and the obstetric nurse's intervention to improve sleep quality and the presence of depressive symptomatology. The total Pittsburgh Sleep Quality Index (PSQI) and Edinburgh Postpartum Depression Scale (EPDS) scores are continuous variables with non-normal distributions. However, these scores were classified into low-risk and high-risk groups, which were used in subsequent non-parametric tests (Mann-Whitney-U test and Kruskal-Wallis test). Non-parametric tests were used to analyze the relationship between individual and clinical characteristics of the sample and depression (EPDS score ≥ 12) and poor sleep quality (PSQI score > 5). Considering the sample size, with non-normal distributions, the Mann-Whitney-U test was performed to identify significant differences in PSQI between two groups: probable depression and no depression. Significance was measured to the $p < 0.05$ level.

## 3. Results

The sample comprises 53 women, with 32 (60.4%) pregnant at 28 weeks or more and 21 (39.6%) giving birth less than seven days ago. They range in age from 18 to 45 years old, 79.2% ($n = 42$) are 35 years old or younger, and 20.8% ($n = 11$) are 36 years old or older. The majority are Portuguese ($n = 44$, 83.0%), with the remaining women being of other nationalities. Of note, 20 (37.7%) were nulliparous. The descriptive analysis of the variables that characterized the sample is shown in Table 1. When the quality of sleep was evaluated, we found that only 9.2% ($n = 9$) had good quality sleep (Table 1). The median PSQI score was 10 (±3.63). In addition to the sleep total score, each of the sleep components was also evaluated, the parts Subjective Sleep Quality (1.68 ± 0.644), Sleep Disturbances (1.74 ± 0.593), and Daytime Dysfunction (2.32 ± 0.581) being the highest scores (Table 1). The median EPDS score was 12 (±4.43), and 27 participants (50.9%) were defined as having probable depression with scores greater than or equal to 12 (Table 1).

**Table 1.** Individual and clinical characteristics of the sample.

| Variable | Number (%) | Median (IQR) |
|---|---|---|
| Pregnancy | 32 (60.4) | |
| Early postpartum period | 21 (39.6) | |
| Age (years) | | |
| 18–30 | 19 (35.8) | |
| 31–35 | 23 (43.4) | |
| ≥36 | 11 (20.8) | |
| Nationality | | |
| Portuguese | 44(83) | |
| Other | 9 (17) | |
| Gestation | | |
| 1 | 20 (37.7) | |
| ≥2 | 33 (62.3) | |
| Sleep measures | | |
| Total PSQI score | | 10 (4–18) |
| Poor sleep quality (>5) | 44 (90.8) | |
| Good sleep quality (≤5) | 9 (9.2) | |
| Subjective Sleep Quality | 53 (100) | 1.68 ± 0.644 |
| Sleep Latency | 53 (100) | 1.53 ± 1.012 |
| Sleep Duration | 53 (100) | 1.17 ± 1.033 |
| Sleep Efficiency | 53 (100) | 1.13 ± 1.301 |
| Sleep disturbances | 53 (100) | 1.74 ± 0.593 |
| Use of Sleep-Promotion Medication | 53 (100) | 0.08 ± 0.432 |
| Daytime Dysfunction | 53 (100) | 2.32 ± 0.581 |
| Depression measures | | |
| Probable depression (≥12) | 27 (50.9) | 18.26 ± 2.31 |
| No depression symptoms (<12) | 26 (49.1) | 5.81 ± 3.13 |

IQR—Interquartile range.

When questioned about the obstetric nurse's intervention to improve sleep quality, 41 (77.4%) participants said that during pregnancy, the obstetric nurse showed no interest in their sleep patterns (Table 2). Moreover, 43 ($n$ = 81.1%) participants reported that during pregnancy, the obstetric nurse showed no interest in their sleep quality (Table 2). The 10 (18.9%) participants who responded affirmatively about the obstetric nurse's interest in their quality of sleep reported that the nurse had addressed the following strategies for improving sleep quality: sleeping positions; the comfort of the environment; sleep schedules; use of comfortable clothing for sleep (Table 2).

The PSQI score was significantly different ($p$ = 0.016) between pregnant and postpartum women with probable depression and no depression symptoms (Table 3). In addition, the pregnant and postpartum women with likely depression had worse sleep quality scores (Table 3).

When performing Mann-Whitney-U and Kruskal-Wallis tests for comparative effects between individual and clinical characteristics and depression (EPDS score ≥ 12) and poor sleep quality (PSQI score > 5), it is found that the value of sleep quality is not significantly different between the individual and clinical characteristics of the participants, especially for the variable's pregnancy period, age, nationality, and gestation ($p$ > 0.05) (Table 4). The same results for the relationship between those variables and probable depression ($p$ > 0.05), except for the variable nationality ($p$ = 0.013) (Table 4). Statistically significant differences were identified between Portuguese women and women of another nationality regarding the probability of depression, with women of another nationality having a higher risk of depression (Table 4).

**Table 2.** Obstetric nurse intervention on sleep quality.

| Nurse Intervention | Number (%) |
|---|---|
| During pregnancy, did the obstetric nurse show interest in your sleep pattern at any time? | |
| Yes | 12 (22.6) |
| No | 41 (77.4) |
| During pregnancy, did the obstetric nurse show interest in your sleep quality at any time? | |
| Yes | 10 (18.9) |
| No | 43 (81.1) |
| If you answered yes, mark all the strategies that were addressed: | |
| Positions to sleep | 6 (11.3) |
| Use of pillows | - |
| The comfort of the environment | 3 (5.7) |
| Use of electronic devices in the bedroom (at bedtime) | - |
| Sleep schedule | 1 (1.9) |
| Physical exercise and its relationship with sleep pattern | - |
| Diet and its relationship with sleep pattern | - |
| Caffeinated beverages | - |
| Muscle relaxation techniques | - |
| Music therapy | - |
| Chromotherapy | - |
| Use of comfortable sleeping clothes | 2 (22.6) |
| Other | - |

**Table 3.** Mann-Whitney-U test between PSQI score and pregnant and postpartum women with probable depression.

| | Depression Measures | | Test | |
|---|---|---|---|---|
| | Probable Depression (*n* = 27) | No Depression Symptoms (*n* = 26) | U de Mann-Whitney | *p* |
| | Mean Rank | | | |
| PSQI | 31.98 | 21.83 | 216.500 | 0.016 * |

* Significance to *p* < 0.05.

**Table 4.** Relationship between individual and clinical characteristics of the sample and depression (EPDS score $\geq$ 12) and poor sleep quality (PSQI score > 5).

| | Probable Depression EPDS Score $\geq$ 12 | | Poor Sleep Quality PSQI Score > 5 | |
|---|---|---|---|---|
| | % (*n/N*) | *p* | % (*n/N*) | *p* |
| Pregancy or Puerperium | | | | |
| Pregnancy | 50.0 (16/32) | 0.867 | 90.6 (29/32) | 0.585 |
| Early postpartum period | 51.4 (11/21) | | 85.7 (18/21) | |
| Age (years) [1] | | | | |
| 18–30 | 57.9 (11/19) | | 84.2 (16/19) | |
| 31–35 | 56.5 (13/23) | 0.403 | 87.0 (20/23) | 0.216 |
| $\geq$36 | 27.3 (3/11 | | 100 (11/11) | |
| Nationality [2] | | | | |
| Portuguese | 43.2 (19/44) | 0.013 * | 86.4 (38/44) | 0.244 |
| Other | 88.9 (8/9) | | 100 (9/9) | |
| Gestation [2] | | | | |
| 1 | 45.0 (9/20) | 0.216 | 90.0 (18/20) | 0.815 |
| $\geq$2 | 54.5 (18/33) | | 87.9 (29/33) | |

* Significance to *p* < 0.05, [1] Kruskal-Wallis test, [2] Mann-Whitney-U test.

## 4. Discussion

Consistent with previous findings [7,13–15,28,29], the participants of this study had poor quality sleep. Commonly, poor sleep quality is associated with physiological changes such as an increased need to urinate, lower back pain, or restless leg syndrome [28]. Furthermore, this interpretation could lead to the devaluation of the poor quality of sleep, either by women and family or by health professionals, assuming that it is normal and physiological in pregnancy.

From the perspective of pregnant women, the present study demonstrated that the obstetric nurse's intervention was insufficient regarding the participants' sleep quality. The strategies addressed by the nurses as promoters of sleep quality were also slightly explored. Sleeping positions, the comfort of the environment, sleep schedules, and use of comfortable clothing for sleep were the strategies addressed by the obstetric nurses. Promoting health and health education is an autonomous intervention and the responsibility of nurses. It is essential that the evidence supports the realization of this health education and, together with women's preferences, that their needs are met. A study of 454 pregnant women recommends that health professionals pay more attention to sleep problems and advise on strategies to improve sleep quality during prenatal examinations [11]. Several studies have proposed methods to promote sleep in pregnant women. Citrus aurantium essential oil efficiently enhances sleep quality in pregnant women. In this study, all the components of the PSQI were significantly enriched in the Citrus aurantium critical oil group compared with the placebo group [29]. The home-based tele-Pilates exercise program is an effective exercise to alleviate anxiety and insomnia experienced by a pregnant woman [30]. In a meta-analysis to understand the effectiveness of non-pharmacological interventions in improving the quality of sleep in pregnant women, they concluded that listening to music, physical exercise, relaxation exercises, lettuce seed, sleep hygiene, and acupressure [13]. All these non-pharmacological interventions are relevant, considering pregnant women are generally cautious about the use of drugs during pregnancy [13]. These interventions corroborate with non-pharmacological and pharmacological treatments considering the risks and benefits of each for the expectant mother and fetus described in a study about insomnia during pregnancy [23]. According to these authors, non-pharmacological interventions such as sleep hygiene and education should be considered as first-line [23].

Assessing sleep quality in the perinatal period is crucial to promoting women's health during the prenatal and postnatal periods [20]. Also, "the quality of night's sleep impacts the subsequent day, and the level of daytime sleepiness impacts the following sleep night" [31], which will be a reality for future mothers due to the need to wake up to take care of the newborn during the night. About one in two women in the present study experienced symptoms of depression. We found significant differences in probable depression and nationality, with women of another nationality having a higher risk of depression. Risk groups are identified in which they are more likely to present sleep disorders, and midwifery should adopt good practices to detect these signs and symptoms. For example, women with low socioeconomic income, obesity, insomnia, and depressive background are more likely to have sleep problems, according to a literature review about the association between sleep quality and perinatal depression [20]. In the same study, it can be found that sleep quality is worse with a maternal age of 30 and over, and is worse with gestational age, third trimester [11–13,20]. A literature review aimed to identify the main barriers and facilitators to dealing with perinatal depression and found that the main obstacles are stigma, lack of training for the obstetrician, lack of resources, and limited access to mental health treatment [32]. Facilitators include validation and empowerment of women during interactions with healthcare providers, midwifery and staff training, standardized screening and referral processes, and enhanced mental health resources [32].

We also found that pregnant and postpartum women with likely depression self-reported poor sleep quality. This finding is consistent with other data suggesting that poor subjective sleep quality is associated with symptoms of perinatal depression [13,33–35]. In addition, Park et al. claim that sleep maintenance may play a more critical role than quantity in developing postpartum mood symptoms [33].

The present study has limitations: the non-use of objective measures to evaluate sleep quality and, whenever possible, actigraphy should be included in the methodology of studies on the subject. Indeed, according to Barbato (2021), subjective instruments for assessing sleep quality did not "focus on sleep components that can lie behind sleep quality and not give cues regarding the biological mechanisms that might be responsible for altering the sleep process and disturbing sleep continuity" [36]. Another limitation is the sample size which does not allow the generalization of the results. The study only analyzes the

women's perspective on the strategies adopted, and future studies should also include the midwives' perspective on their role in promoting sleep quality. Furthermore, the probability of the previously validated cut-off score of five In the PSQI is not correct for the pregnant population due to changes that occur during pregnancy [37]. A higher score may be needed to differentiate those who need further intervention [37]. In the same study, a meta-analysis of sleep quality during pregnancy, suggests that higher global score cut-off points are more appropriate [37]. However, there is no consensus about the score for this population. So, future research, including mixed methodologies, must be conducted to identify the best cut-off score for the pregnant population. A meta-analysis of non-pharmacological interventions for improving sleep quality during pregnancy included studies between July 2014 and July 2019; only six were included [25]. Thus, randomized controlled trials are needed to understand the effectiveness of non-pharmacological interventions in promoting sleep quality. However, it is possible to identify that the women in this study have poor sleep quality, and that they perceive common concern for their quality of sleep on the part of the midwives. This suggests the need for more action since we are facing one of the most significant challenges of this century, preventing depression.

## 5. Conclusions

Women's mental health during pregnancy and postpartum impacts them, as well as the bond created with the child and the family dynamics. Sleep is a state of relaxation necessary for all human beings. Indeed, the interaction between sleep and depressive symptoms is quite complex because we can always ask: What causes what? Does quality of sleep cause depression? Or is the poor quality of sleep a symptom of depression? The evidence shows that it is difficult to understand this complexity of interplay. Still, it also indicates that non-pharmacological interventions promote sleep quality and reduce the risk of postpartum depression. Therefore, the obstetric nurse must focus on each woman's needs to identify a plan that provides an optimal level of sleep. In addition, poor sleep quality during pregnancy increases the risk of perinatal depression, which harms the woman, child, and the rest of the family. More studies with a longitudinal approach should understand the impact of interventions and the best interventions of the obstetric nurse to improve sleep quality, considering that obstetric nurses are the health professionals closest to pregnant women. They also assume a role in monitoring the health of pregnant women and act autonomously in health promotion and risk prevention, being able to fit into their plan of diagnostic care and nursing interventions that respond to the need for sleep and rest of pregnant and puerperal women.

**Author Contributions:** Conceptualization, A.F.P. and M.O.Z.; methodology, A.F.P. and M.O.Z.; formal analysis, A.F.P.; investigation, A.F.P. and M.O.Z.; data curation, A.F.P.; writing—original draft preparation, A.F.P.; writing—review and editing, A.F.P. and M.O.Z.; project administration, A.F.P. and M.O.Z.; Funding acquisition, A.F.P. All authors have read and agreed to the published version of the manuscript.

**Funding:** This research received no external funding.

**Institutional Review Board Statement:** This study was approved by Hospital's Ethics Committee (approval number: 47/2021). The authors do not mention the ethics committee's name to maintain the anonymity of the participants since the ethics committee is from the hospital where the data were collected. The ethics committee's opinion files were made available to the journal's editorial board.

**Informed Consent Statement:** Not applicable.

**Data Availability Statement:** Not applicable.

**Conflicts of Interest:** The authors declare no conflict of interest.

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
