# Peer review of "Improving Sleep Quality to Prevent Perinatal Depression: The Obstetric Nurse Intervention"

_2673-8937, doi:10.3390/ijtm3010004_

Round 1
Reviewer 1 Report
- Dear Authors and Editors,
- I would like to express my great respect for reading very important and well organized article, which answers the original and well-defined questions.
- I would present main possitive aspects of the text:
- The abstract - is short but coherent outlining the aim of the reserach, methods, results and main contributions
- The article body is also well organised, firstly the main topic is introdused based on the modern and adequate literature search, secondly the material and method are described. Presented study based on the quantitative and cross-sectional paradigm. In reference to the given hypothesis no control group was needed. The main criterion was the depression assessment to separate two subgroups: with probable depression vs with no depression symptoms.
- Results are presented clearly in the reference to the main aim: to find the relationship between depression and poor sleep quality as well to the analysis the individual and clinical characteristics influence. The adequate statistical methods were used to find find significant differences mentiond above two subgroups. The other important aim was to find whether the obstetric nurse showed interest in sleep problems of pragnant woment to improve sleep quality.
- Results are described very clearlly with all details. All important results are presented in four tables.
- Discussion extensively analyzes the relationships between the data and interprets them with reference to the cited literature. I would like to draw attention to the result obtained by the Authors that nurses do not pay attention to sleep problems of pragnant women. This is very worrying and puzzling because of the importance of sleep to the health and well-being of pregnant women and the prenatal (and postnatal) development of their babies.
- In the article conclusions, I would suggest to point out more strongly the need to sensitize the staff to these problems and prevent somatic and mental health by improving the quality of sleep patterns in pregnant women.
- To sum up I would like to express that: the manuscript is clear, relevant for the field and presented in a well-structured manner. The cited references are current and relevant. The only one self-citations is refered. The manuscript is good quality and presented experimental design appropriate to answer the research questions. The methods section is fool of adequate details. All four tables are appropriate and the data are properly showed and easy to interpret and understand. The results based on the adequate statistic instruments are interpreted appropriately. Ethics statement is mentioned as well as the limitations and conclusions
Author Response
Dear Reviewer,
We appreciate your kind comments. It was very gratifying to see your excellence recognized for all the effort and work necessary to arrive at this version of the article.
We accept your suggestion and thank you, and we improve the conclusions:
They also assume a role in monitoring the health of pregnant women and act autonomously in health promotion and risk prevention, being able to fit into their plan of diagnostic care and nursing interventions that respond to the need for Sleep and Rest of pregnant and puerperal women.
Best regards
Reviewer 2 Report
Dear Authors/Editor,
Thanks for the opportunity to review your manuscript. I think the topic is relevant, the research design is strong and the manuscript is clear and well written. Here are some minor comments and questions:
First, Key words: MeSH indexed key words should be used. Order them alphabetically.The sentence regarding the Ethics Committee approval should be placed on the beginning of the subjects and methods section.
Please clarify how did the authors obtain informed consent from participants of this
suryey.The sample is small and could be better characterised.Questionnaire on the socio-economic status of the pregnant women please explain which variables were used to describe the socioeconomic status of a participant. Did you used questionnaire that was used in previous surveys or was it constructed for the present research?Which is the reason to examine pregnant women from 28 weeks of pregnancy and the 7th day after delivery.Is there a special jump in hormonal values ​​during this period that affect the mood of pregnant women and mothers in labor?Can the family of the woman in labor help the woman in labor not have a sleeping problem or is it just the nurse's scope of work.To what extent was the nurse involved in educating the family of the mother and pregnant woman for better sleep and does she have an impact on the family of the pregnant woman!Is there a difference in the quality of sleep of pregnant women and women in labor, and if so, can you explain it.Please use more recent references!
Author Response
Dear Reviewer,
Thank you for your time and comments to improve this article.
We present the changes and responses to comments in table format in the attached document.
Best regards,

Reviewer 3 Report
Dear authors.
Thank you for this interesting topic you touch.
Some concerns to check or improve manuscript:
1) Methods / inclusion criteria: too many repetitions of "pregnant women" . You may say "pregnant woment plus other scriteria in order not to be too repetitive.
2) It is not surprising thta people of other natinalities could have more risk to depression since there are many studies reporting higher risks of mental health amoing immigrants.
With best regards,
Liudmila
Author Response
Dear Reviewer,
Thank you for your time and comments to improve this article.
We accept your suggestion, and we have improved the inclusion criteria:
Inclusion criteria were cumulative: Pregnant women 1) between the 28th week of pregnancy and the 7th day after delivery; 2) who understand and can express themselves in Portuguese; 3) who have access to and can use a computer or smartphone with an internet connection.
Thank you one more time.
Best regards